# Miniaturized Metamaterial-Inspired Travelling Wave Tube for S Band

**Ying Xiong ¹,\*, Xianfeng Tang ²,\*, Juncheng Ma ¹ and Liping Yu ¹**

¹ School of Engineering and Technology, Chengdu College of University of Electronic Science and Technology of China (UESTC), Chengdu 611731, China; majuncheng@cduestc.edu.cn (J.M.); woodbeauty@163.com (L.Y.)

² School of Physical Science and Technology, Southwest Jiaotong University (SWJTU), Chengdu 611756, China

\* Correspondence: scxiongying@sina.com (Y.X.); tangxianfeng@swjtu.edu.cn (X.T.); Tel.: +86-15184432176 (Y.X.); +86-13568957994 (X.T.)

**Abstract:** A miniaturized traveling wave tube (TWT) was studied by proposing a novel metamaterial (MTM) slow wave structure (SWS). The dispersion results show that $n = -1$ space harmonic of the fundamental mode exhibits the "forward" wave properties, which is the foundation of the MTM-inspired TWT. Meanwhile, the interaction impedance for mode 2 of the novel MTM SWS can be sharply decreased by introducing four blend edges to weaken the corresponding longitudinal electric field. Also, two coaxial couplers are presented to input/output the signals. The transmission results show that the reflection is as low as $-15$ dB from 2.90 GHz to 3 GHz, which ensures the amplified signal can be effectively outputted. The MTM-inspired TWT exhibits miniaturized superiority for its compact high frequency structure including the MTM SWS and the coaxial couplers. Especially, for the high-frequency structure, the transverse and longitudinal sizes are $\sim\lambda/5$ and $\sim3\lambda$, respectively ($\lambda$ is the free-space wavelength at the operating frequencies). The simulation of the beam wave interaction shows that the proposed MTM-inspired TWT yields output powers of kW levels from 2.90 GHz to 3 GHz, with a gain of 23.5–25.8 dB and electronic efficiency of 14–22% when the beam current is 0.5 A and the beam voltage is 13 kV. The results indicate that the gain per wavelength is as high as 8.5 dB in the operating bands. The simulation results confirm that it is possible to weaken the backward wave oscillation from the higher mode in the miniaturized MTM-inspired TWT.

**Keywords:** metamaterial; travelling wave amplifier; miniaturization; coaxial coupler; sheet electron beam



## 1. Introduction

Traveling wave tubes (TWTs) are one of the most important vacuum electron devices (VEDs), in which the slow-wave structures (SWSs) are the key components and could greatly affect the device's performance. Up to now, many SWSs such as the helix, the coupled cavities, the staggered double vanes, the folded waveguides, the sine waveguides, et al. are presented to improve the performance of TWTs including but not limited to high power, wide bandwidth, high electronic efficiency, or high gain, et al. [1–7]. For example, the helix TWTs have the ultra-wideband though their power capacity is limited by the dielectric support rods of the SWSs. The coupled-cavity TWTs have narrow bandwidths but exhibit high output powers. The other TWTs based on the staggered double vanes, folded waveguides, and sine waveguides have trade-offs between the power and bandwidth but their sizes are large, especially in the microwave bands. As a result, different SWSs can offer different advantages for the TWTs, which contribute to their wide applications in the fields such as high-date rate communications, navigation satellites, high-resolution radars, et al. [8–10]. All the time, for the TWTs or the other VEDs, miniaturization is one of the important and long-term objectives, which means smaller sizes and lighter weight and thus is beneficial for the improvement of their competitiveness.

Recently, metamaterials (MTMs) have attracted much attention to VEDs for exotic electromagnetic characteristics such as reversed Cherenkov radiation (RCR) effect [11–14]. The MTMs can be used to construct the novel SWSs based on the RCR. This is because the longitudinal electronic fields in the MTMs can interact with the electron beam and thus be able to generate coherent radiation. Meanwhile, the MTM SWSs have the advantages of miniaturization and high interaction impedances. The miniaturization characteristics are derived from the subwavelength properties of the MTMs while the high interaction impedances result from the enhanced longitudinal electric fields in the MTMs. The former is in accordance with the developed trend of VEDs and the latter ensures the high output power of the MTM-inspired VEDs.

At present, the MTMs have been successfully used to realize the backward wave oscillators (BWOs), the extended interaction klystrons, and the THz TWTs [15–20]. In the MTM-inspired THz TWTs, the waveguide couplers are used to input and output the signals [20]. However, when the MTMs are used to design S band TWTs, the waveguide couplers would enlarge the total sizes, which is not beneficial for miniaturization. In our previous work, the MTM-coaxial coupling technology [21] is presented in the MTM-inspired BWO to realize miniaturization. Especially, the MTM has two modes with high interaction impedances, which are derived from the enhanced longitudinal electric fields. The property is beneficial for BWOs, which can be used to construct dual-band BWO [22]. Nevertheless, when this MTM is applied to the TWT, the above property would lead to the strong backward wave oscillation from the higher mode and could greatly degrade the performance of the TWT.

In this paper, in order to suppress the backward wave oscillation from the higher mode, a novel MTM SWS is proposed by using the High-Frequency Structure Simulator (HFSS) of ANSYS Electronics Desktop [23] to analyze the longitudinal electric field distribution and interaction impedances. Also, the MTM-coaxial coupling technology is introduced to construct the MTM-inspired TWT based on the novel MTM SWS. Finally, the beam-wave interaction simulations of the MTM-inspired TWT are performed by using the Particle Studio of Computer Simulation Technology (CST) [24].

## 2. Analysis of the MTM SWS with the Input and Output Couplers

In our previous work, an MTM SWS was presented to develop the BWO. The unit cell of the initial MTM SWS is shown in Figure 1a, which consists of the hollow square waveguide and its loaded complementary electric split ring resonator (CeSRR) [25]. The dispersion curves for the initial MTM SWS are obtained by using the eigenmode solver of HFSS [24], as shown in Figure 1b. When the beam line intersects with mode 1 at the 0st spatial harmonic or "backward" wave area, the MTM SWS can be used to construct the BWO [22]. Here, in order to develop the MTM-inspired TWT based on the SWS, the beamline should intersect with mode 1 at −1st spatial harmonic or "forward" wave behavior. However, as shown in Figure 1b, the beamline of ~12 kV not only intersects with mode 1 at ~2.89 GHz but also with mode 2 at ~4.04 GHz.

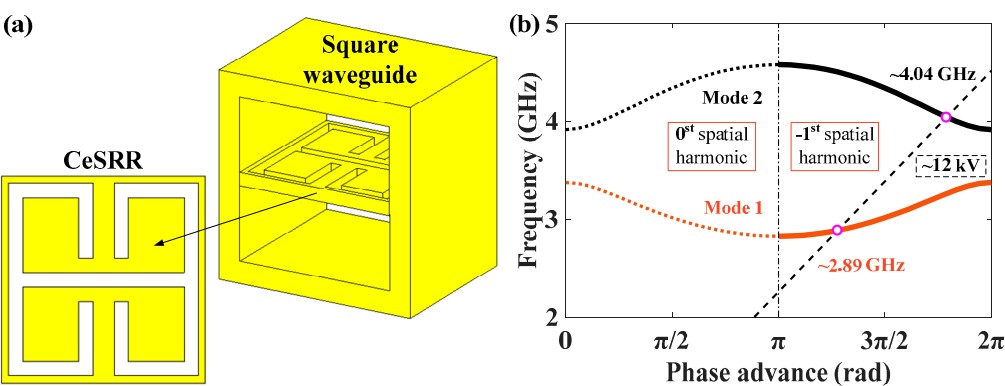

**Figure 1.** (**a**) The unit cell of the initial MTM SWS and (**b**) its dispersion curves.

Meanwhile, the electric fields of the two modes are shown in Figure 2. From the point of the electric field, the longitudinal electric field of mode 1 distributes in the center of the CeSRR, which is used to interact with the sheet electron beam and thus contributes to the amplification of the signal. Also, the longitudinal electric field of mode 2 distributes around the corner of the CeSRR, which can interact with the sheet electron beam to generate the backward wave signal and destroy the amplification of the signal of mode 1. Hence, in order to improve the performance of the MTM-inspired TWT, it is necessary to decrease the longitudinal electric field of mode 2 and thus suppress the unexpected backward wave oscillation from mode 2.

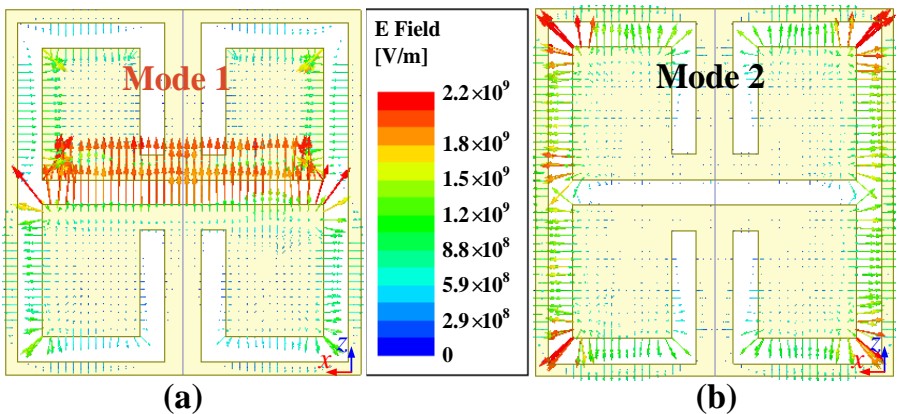

**Figure 2.** Electric field distribution of (**a**) mode 1 and (**b**) mode 2 for the initial MTM SWS.

According to the above analysis of the electric field, we propose the novel MTM SWS, as shown in Figure 3a. The novel MTM SWS includes the square waveguide and the novel CeSRR. It should be noted that the novel CeSRR is formed by introducing four blend edges with a radius $R = 4$ mm on the basis of the initial CeSRR. On one hand, the dispersion curves of the novel MTM SWS are obtained by using the eigenmode solver of HFSS [23], as shown in Figure 3b. Here, in order to compare with the initial MTM SWS, the same operating point (or the same phase advance) is selected for the novel MTM SWS. As a result, for the novel MTM SWS, the beam voltage line of ~13 kV intersects with mode 1 and mode 2 at ~2.92 GHz and ~4.30 GHz, respectively. Meanwhile, the electric fields for mode 1 and mode 2 of the novel MTM SWS are shown in Figure 4a,b, respectively. It is seen that the electric field distribution of mode 1 is almost unchanged while the electric field distribution of mode 2 almost rotates 90° due to the four blend edges. Compared with the longitudinal electric field of mode 2 in Figure 2b, the longitudinal electric field of mode 2 for the novel MTM SWS sharply decreases.

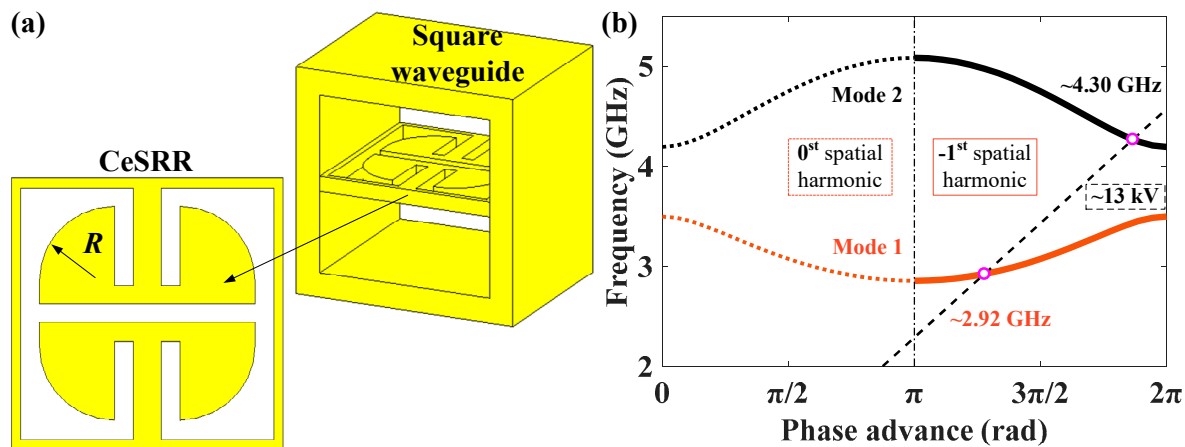

**Figure 3.** (**a**) The unit cell of the novel MTM SWS and (**b**) its dispersion curves.

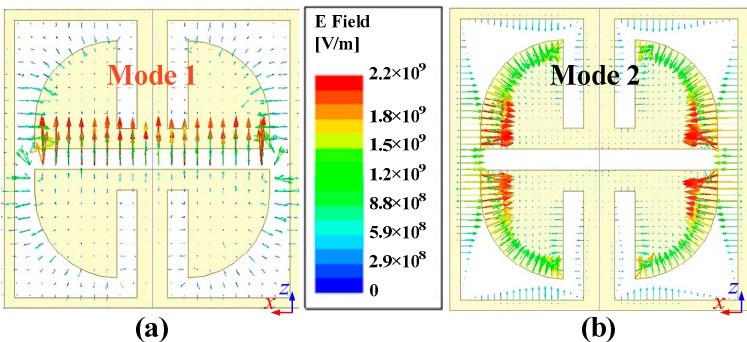

**Figure 4.** Electric field distribution of (**a**) mode 1 and (**b**) mode 2 for the novel MTM SWS.

Further, in order to quantitatively describe the advantage of the novel MTM SWS with respect to the initial one, we calculate the interaction impedances $K_c$ of the novel and the initial MTM SWSs by using the field calculator of HFSS [24]. $K_c$ is an important parameter of the SWS, which is written as

$$K_c = \frac{|E_{zn}|^2}{2\beta_n^2 P_w} \tag{1}$$

where $|E_{zn}|$ is the longitudinal electric field amplitude of the *n*-th space harmonic, $\beta_n$ is the phase constant, and $P_w$ is the power flow along the transportation direction of the sheet electron beam [22]. Here, a higher longitudinal electric field leads to higher $K_c$. Meanwhile, higher $K_c$ means more intense backward wave oscillation [25]. That is why decreasing the longitudinal electric field could suppress the backward wave oscillation.

The interaction impedances of the novel MTM SWS and the initial one are shown in Figure 5a,b respectively. It is clear that for mode 1, the interaction impedances of the two MTM SWSs are almost the same. This is because the longitudinal electric field amplitude of mode 1 is almost unchanged with introducing into the four blend edges. As a comparison, for mode 2, the interaction impedance of the novel MTM SWS is 1/3 lower than that of the initial one. Compared with initial MTM SWS, four blend edges for the novel MTM SWS lead to the decrease of the longitudinal electric field of mode 2 and thus lower the interaction impedance of the novel MTM SWS. The facts indicate that the novel MTM SWS has lower interaction impedance and thus the backward wave signal from mode 2 has lower power with respect to the initial one.

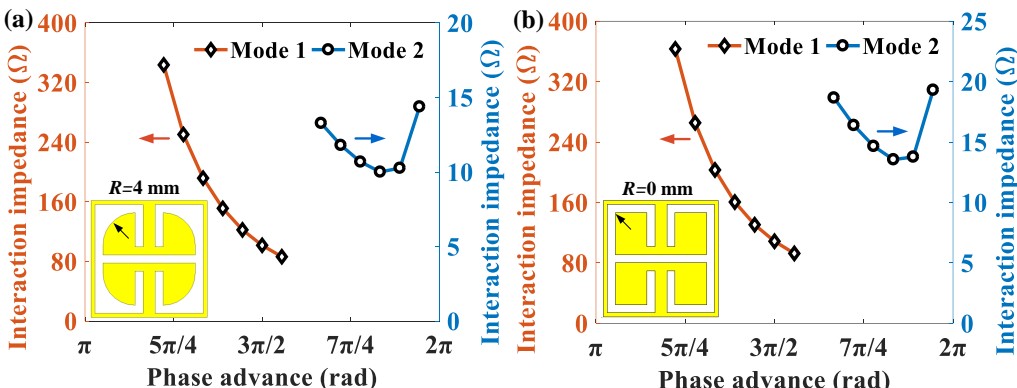

**Figure 5.** Interaction impedances for (**a**) the novel MTM SWS and (**b**) the initial MTM SWS.

The MTM SWS offers the precondition for the signal amplification of mode 1. Meanwhile, the input coupler and output coupler should be designed to couple the input signal and output the amplified signal, respectively. Here based on the MTM-coaxial coupling technology [21], we design the input and output coaxial couplers and then build the transmission model of the MTM-inspired TWT. The simulation model is in Figure 6. It is found that the simulation model consists of one input coaxial coupler, one output coaxial coupler,

and the MTM SWS with 18 periods. The input (or output) coaxial coupler consists of a 50 Ω Sub Miniature version A (SMA) receptacle, a matched block, and a CeSRR, in which the inner conductor of the SMA is fixed on the matched block. Also, the SMA with 1.3 mm inner diameter and 4.2 mm external diameter is filled with Teflon. The transmission simulation is performed by using the Microwave Studio of CST [24]. In the simulation, the copper conductivity is set as $5.8 \times 10^7$ S/m. The transmission and reflection results are shown in Figure 7. It is seen that the $|S_{21}|$ is −1 dB~−2 dB and the $|S_{11}|$ is less than −15 dB from 2.9 GHz to 3 GHz, indicating that the novel MTM SWS also exhibits good transmission characteristics and low reflection with respect to the initial one [21]. As a result, the coaxial coupler is suitable for the MTM-inspired TWT to input and output the signals.

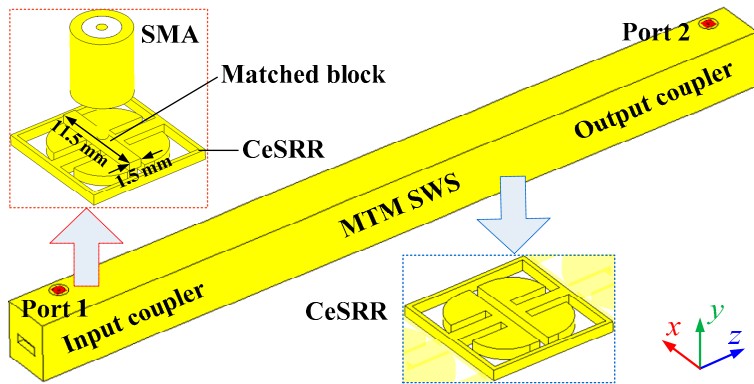

**Figure 6.** The transmission model of the MTM-inspired TWT. (Inset: the input and output couplers and the novel MTM SWS).

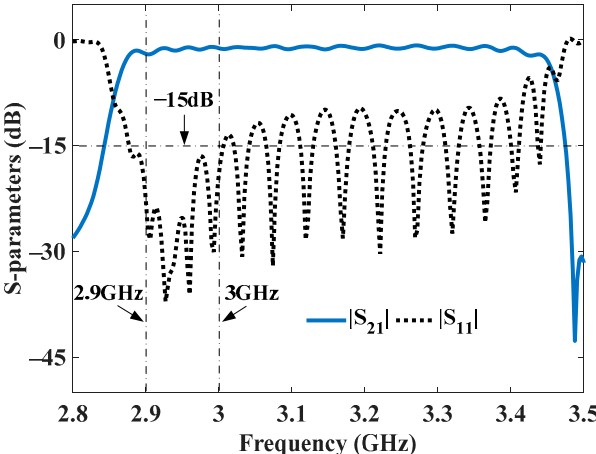

**Figure 7.** S-parameters vs. frequency.

## 3. Analysis of the Beam-Wave Interactions

According to the analysis of the MTM SWS and the input/output coaxial coupler, we build the interaction simulation model shown in Figure 8 and then carry out the beam-wave interaction simulations of the MTM-inspired TWT by using the particle-in-cell (PIC) solver in the Particle Studio of CST [24]. As we see, the interaction model is built by adding the cathode to the transmission model. The cathode, which is used to emit the sheet electron beam, has a width of 8 mm, a height of 2 mm. Also, the cathode is located on the opposite side of the coaxial couplers and the space between the cathode and the CeSRRs is 0.5 mm. According to the dispersion in Figure 3b, the beam voltage is set as 13 kV to ensure the beamline can intersect with mode 1 at the backward wave area. Also, the beam current of the sheet electron beam is set as 0.5 A, which is obtained by scanning the beam current, to drive the MTM-inspired TWT and thus to amplify the input signal of 3 W (or ~2.45 V) at

2.95 GHz. Meanwhile, the uniform axial magnetic field of 0.1 T is used to confine the sheet electron beam.

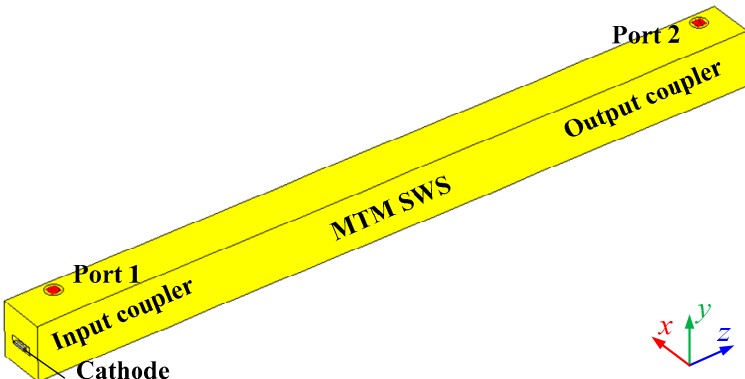

**Figure 8.** Simulation model of the novel MTM-inspired TWT for the beam-wave interaction.

As a result, the voltage amplitudes of the input signal and output signals at port 1 and port 2 of the MTM-inspired TWT as a function of time are shown in Figure 9a. Here, the output signal at port 1 is the reflected signal and the output signal at port 2 is the amplified signal. As we see, the input signal is gradually amplified from port 1 to port 2 and finally the output signal at port 2 becomes stable to ~47 V after 40 ns. Meanwhile, the reflection signal is very small, which ensures the stability of the output signal at port 2. Then the power as a function of frequency is obtained by Fourier transform, as shown in Figure 9b. The results indicate that the output signal at port 2 has the same frequency of 2.95 GHz as the input signal. The input power is ~4.8 dBW and the power of the output signal at port 2 is ~30.6 dBW, which indicates the gain at 2.95 GHz is ~25.8 dB. Meanwhile, the signal at port 1 has a very small power around 4.3 GHz, which is the intersection point between the beam line and mode 2. The result confirms that the backward wave of mode 2 is not excited.

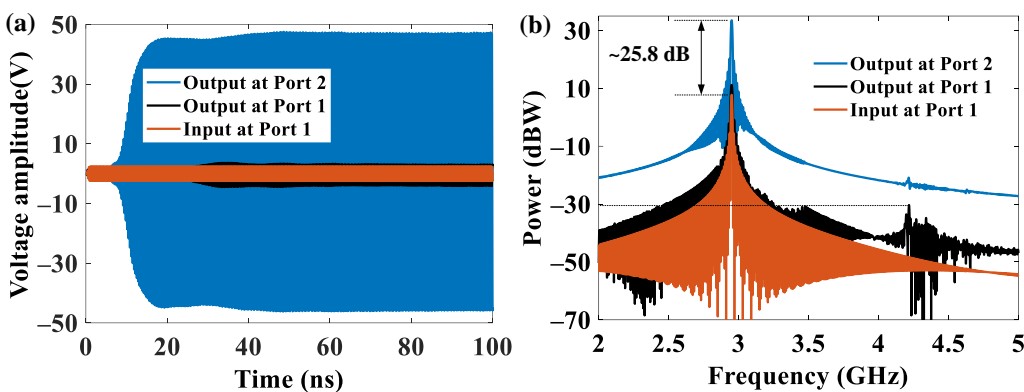

**Figure 9.** (**a**) Voltage amplitudes of input and output signals vs. time and for (**b**) Input and output powers vs. frequency for the novel MTM-inspired TWT.

As a comparison, the beam wave interaction results for the TWT based on the initial MTM SWS are shown in Figure 10. In the simulation, the beam voltage is set as 12 kV according to the dispersion curve in Figure 1b and the other parameters are consistent with those of the novel MTM-inspired TWT. As seen in Figure 10a, the voltage becomes unstable after 60 ns. This phenomenon is derived from the strong backward wave at ~4.04 GHz, which is the intersection point between the electron beam line and mode 2, as shown in Figure 1b. Compared with the beam-wave interaction results between Figures 9 and 10, it is found that the power at ~4.30 GHz for the novel MTM TWT is ~13 dB lower than the one at ~4.04 GHz for the initial one. The results indicate that the backward wave oscillation from mode 2 is suppressed in the MTM-inspired TWT based on the novel MTM SWS for the weaker longitudinal electric field or lower interaction impedance.

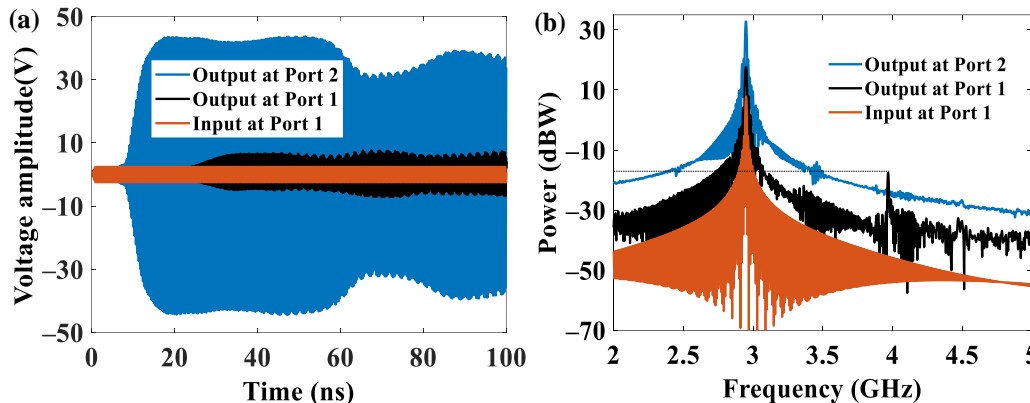

**Figure 10.** (**a**) Voltage amplitudes of input and output signals vs. time and for (**b**) Input and output powers vs. frequency for the initial MTM-inspired TWT.

Furthermore, the electron trajectory of the sheet electron beam for the novel MTM-inspired TWT is shown in Figure 11a. Here, when the input signal transports from port 1 to port 2, the sheet electron beam is well confined by the uniform axial magnetic field of 0.1 T, which ensures the interaction between the sheet electron beam and mode 1. Meanwhile, the sheet beam is gradually and stably modulated by the input signal and thus the input signal obtains energies from the sheet electron beam. Furthermore, the phase space plot at the steady state of 100 ns is shown in Figure 11b. The results show that most of the electrons in the sheet electron beam are slowed down and then the energies of the sheet electron beam are transferred to the input signal to amplify the input signal.

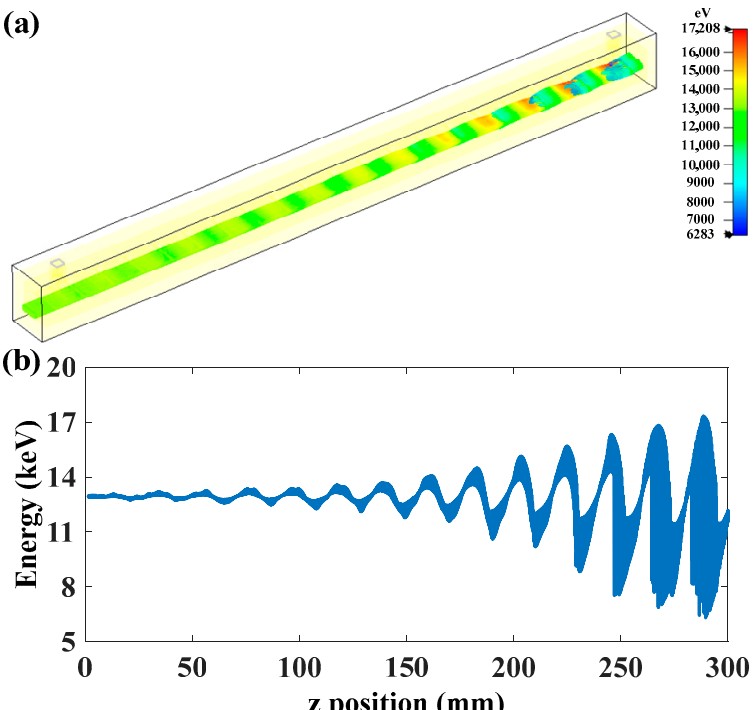

**Figure 11.** Sheet electron beam at 100 ns. (**a**) Electron trajectories and (**b**) phase space plot.

Furthermore, we study the performance of the novel MTM-inspired TWT at different beam currents, different input powers, and different frequencies when other parameters keep unchanged. First, when the input signal is 3 W at 2.95 GHz, the output powers and the gains at different beam currents are shown in Figure 12a. It is found that when the beam current ranges from 0.2 A to 0.7 A, the proposed TWT yields output powers from 280 W to 1400 W, with the corresponding gains from 19 dB to 26 dB. Second, when the beam current

is 0.5 A, the output powers and the gains at different input powers are shown in Figure 12b. It is seen that the output power first increases and then decreases with the increase of the input signal and it achieves the maximum at an input signal of ~3 W. Finally, when the beam current and the input power are 0.5 A and 3 W respectively, the output powers and gains as a function of frequency are shown in Figure 13a. It is clear that the output signal at port 2 has an output power of 600 W to 1100 W from 2.90 to 3 GHz. Meanwhile, the corresponding electronic efficiency is shown in Figure 13b. It clearly shows that electronic efficiency is more than 14% and achieves a maximum of ~22% at 2.95 GHz.

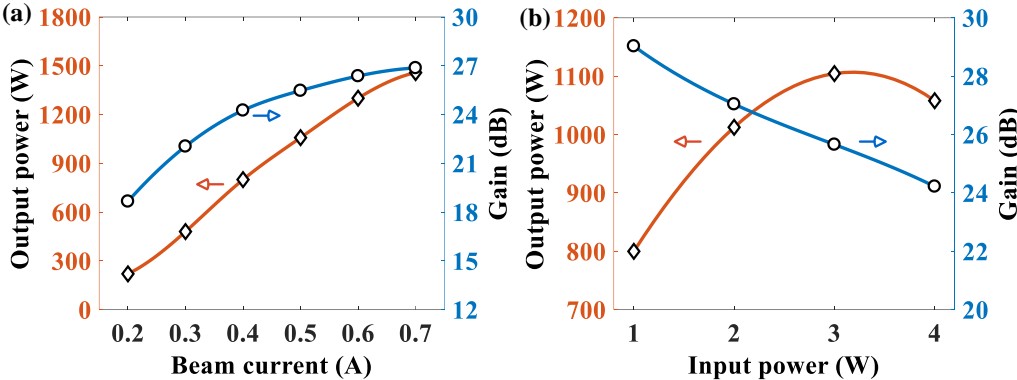

**Figure 12.** (**a**) Output power and gain vs. beam current at 3 W input power. (**b**) Output power and gain vs. input power at 0.5 A beam current.

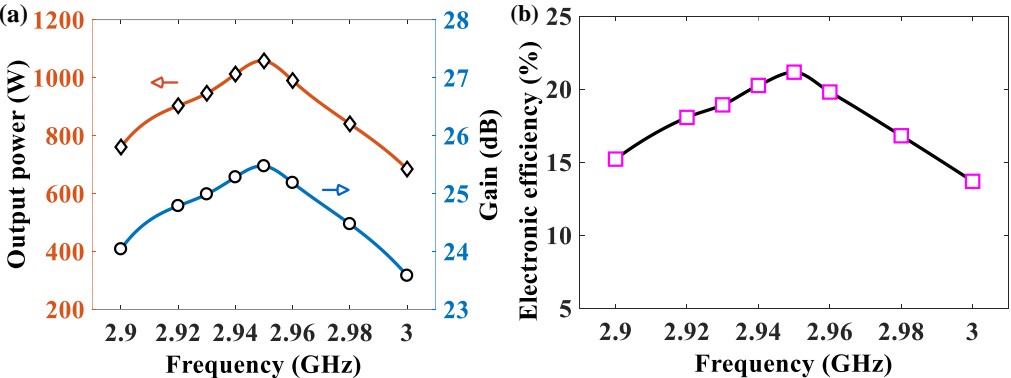

**Figure 13.** (**a**) Output power, gain and (**b**) electronic efficiency as a function of frequency at 0.5 A beam current and 3 W input signal.

## 4. Discussion

In order to discuss the effect of the blend edges on the performance of the MTM-inspired TWT, we further carry out the beam-wave interaction simulations at different *R*. Figure 14 shows the interaction impedances of 2.95 GHz and 4.30 GHz at different *R*. It is clear that the interaction impedances at the two frequencies gradually decrease with the increasing of *R*. Further, the beam wave interaction results for different *R* are shown in Figure 15. Here, it is noted that the results are obtained at 13 kV beam voltage, 0.5 A beam current, and 3 W input signal at 2.95 GHz. As we see, the output power at port 1 has the lower power around 4.30 GHz and the output power at port 2 reaches the maximum at *R* = 4 mm where the output power at port 2 realizes the trade-off. This is because the increasing of *R* would simultaneously lead to the decrease of the interaction impedances at 2.95 GHz and 4.30 GHz. The former might lower the output power at port 2. The latter would contribute to suppressing the backward wave oscillation from mode 2, which is in turn to improve the output power at port 2.

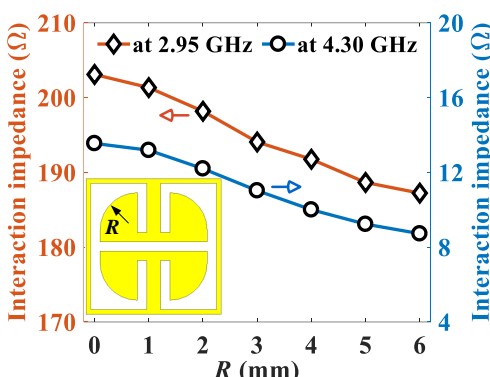

**Figure 14.** Interaction impedances for different *R* at 2.95 GHz and 4.30 GHz.

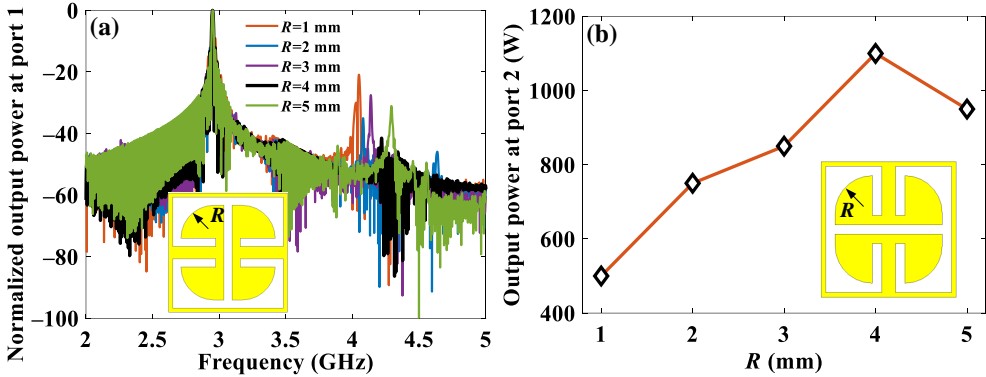

**Figure 15.** (**a**) Normalized output powers at port 1 and (**b**) output powers at port 2 for different *R*.

Furthermore, the results for the blend edges of *R* = 4 mm (case A) and the chamfer edges of *R* = 4 mm (case B) are presented in Figures 16 and 17. Figure 16 shows the interaction impedances for the two cases. It is seen that the MTM SWS with blend edges has higher interaction impedance with respect to the one with chamfer edges. Further, the normalized output powers at port 1 for the two cases are shown in Figure 17a. Also, both of the two cases have a lower output power at port 1 around 4.30 GHz, which indicates that the backward wave signals from mode 2 are almost not excited. Figure 17b shows the output powers at port 2 for the two cases. We see that the MTM-Inspired TWT with blend edges has higher output power compared with the one with chamfer edges.

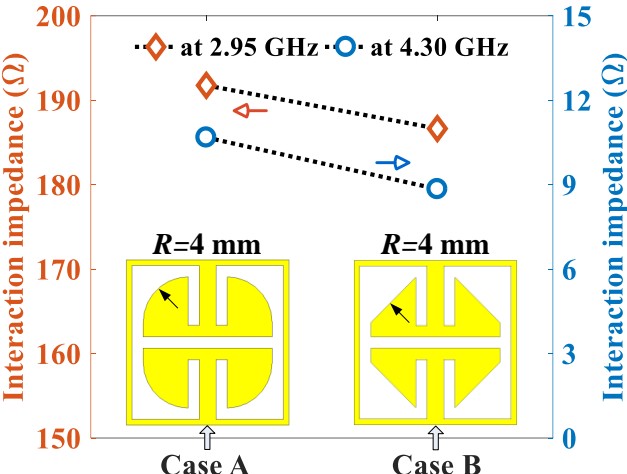

**Figure 16.** Interaction impedances for the blend edges of *R* = 4 mm and the chamfer edges of *R* = 4 mm.

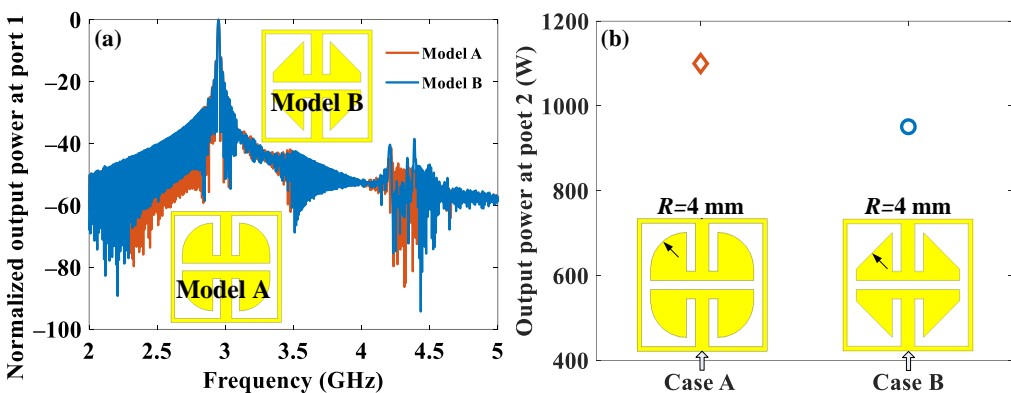

**Figure 17.** (**a**) Normalized output power at port 1 and (**b**) output power at port 2 for different *R*.

The novel MTM SWS is proposed by introducing four blend edges based on the initial MTM SWS, which exhibits a weaker longitudinal electric field and lower interaction impedance for the higher mode (mode 2). Compared with the results at different blend edges, it is clear that the smaller *R* doesn't always mean a higher output power. When *R* achieves a certain value, that is 4 mm, the backward wave signal almost has been completely suppressed. If we continue to increase *R*, the interaction impedance of mode 1 would correspondingly decrease and thus lower the output power. Here, we offer a way to lower the backward wave oscillation from the higher mode in MTM, which is beneficial for the development of the MTM-inspired TWT.

## 5. Conclusions

In this paper, a novel MTM SWS is presented and then a miniaturized TWT is developed based on it. The proposed MTM-inspired TWT exhibits miniaturized properties due to the sub-wavelength of the MTM as well as the compact coaxial couplers. Importantly, comparing the novel MTM SWS with the initial one, it is found that the longitudinal electric field and thus the interaction impedance of the higher mode can be sharply reduced by introducing four blend edges to change the CeSRR. Especially, the beam-wave interactions of the MTM-inspired TWT are carried out by using the Particle Studio of CST. The results show that the MTM-inspired TWT yields output powers of kW levels, with a gain of 25.8 dB in the longitudinal length of ~3λ when the input signal of 3 W at 2.95 GHz. It is feasible that the performance of the MTM-inspired TWT with miniaturization can be improved by suppressing the backward wave oscillation from the higher mode. In the near future, we will carry out the experiment test.

**Author Contributions:** Conceptualization, Y.X. and X.T.; methodology, X.T. and Y.X.; software, X.T.; formal analysis, Y.X.; data curation, J.M. and L.Y.; writing—original draft preparation, X.T.; writing—review and editing, Y.X. and X.T. All authors have read and agreed to the published version of the manuscript.

**Funding:** This work was supported in part by the Fundamental Research Funds for the Central Universities under Grant no 2682021CX072, the Sichuan key research and development program under Grant no 2022YFG0226.

**Data Availability Statement:** The data that support the findings of this study are available from the corresponding author upon reasonable request. The data are not publicly available as they are also part of an ongoing study.

**Conflicts of Interest:** The authors declare no conflict of interest.

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
