# Peer review of "Miniaturized Metamaterial-Inspired Travelling Wave Tube for S Band"

_electronics, doi:10.3390/electronics12143062_

Round 1
Reviewer 1 Report
The paper presents the miniaturized travelling wave tube (TWT) is studied by proposing a novel metamaterial (MTM) slow wave structure (SWS). The introduction contains an overview of current reports related to the research topic. A fairly extensive numerical analysis of the researched issue was presented. As a weakness of the article, I think that there is a lack of experimental studies that are supported by numerical analysis. The conclusions show that the longitudinal electric field and thus the impedance of the higher mode can be reduced by introducing four rounded edges to change the CeSRR.
Author Response
Dear Editor & Reviewer
Please see the attachment, thank you.

Reviewer 2 Report
Page 2. The software used for the simulation was not specified. On which computer was it performed?
Page 3. Line 102. How was the radius of edge rounding determined? Is the value R=4 mm a result of calculations?
Page 9. Discussion. In my opinion, this chapter is incomplete. How would the simulation results change if the rounding radius were 3 mm (or 5 mm)? The authors assumed beforehand that the proposed solution is suitable without justifying their choice.
Page 10. References. Could the authors provide more details about the CST STUDIO SUITE software, which was listed as the last item? Such information should be provided before describing the simulation results.
Author Response
Dear Editor & Reviewer:
Please see the attachment, thank you.

Reviewer 3 Report
This manuscript proposes a metamaterial slow wave structure with a round edge structure on the CeSRR, which provides the effect of suppressing backward wave of higher mode and it has been verified with simulation. It can be considered as quite a unique structure, but lack of hardware verification and in-depth physical explanation.
The round edge of CeSRR would be the main contribution of this manuscript. But this idea looks jumping out without any solid backup from physics. The simulation has shown that the impedence for mode 2 has been largely reduced by introducing this round edge. Would it do the job by using a 45 degree straight cut on the edge? Would it be even better with a radius of 2mm rather than 4mm as specified in the manuscript? It is suggested to either make a prototype to demonstrate the technical breakthrough made by the proposed structure even if it looks like being obtained by accident, or put more reasoning and argument to highlight the physical concept behind this MTM structure.
Quite a lot typos and grammatical errors scattering over the whole manuscript, which needs an extensive editing.
Author Response

(The authors gave the same response as above.)

Reviewer 4 Report
This paper proposed metamaterial traveling wave tube with small profile. This proposed design is the improved version of authors’ previous work that show the improvement in the suppression of backward wave oscillation from higher mode. The performance from simulation results are very interesting and promising. This work is in good term for publications with few small comment as:
- Line 140, there’s a duplicated text on S21 results
- Figure 7, it would be clearer if the authors put the text and state the band 2.9-3GHz in the figure (instead of highlight with lines)
Author Response

(The authors gave the same response as above.)

Round 2
Reviewer 3 Report
The authors have addressed the issues raised in the reviewing comment, and this manuscript can be considered for the publication in Electronics.
N/A